# A Qualitative Study of Child Nutrition and Oral Health in El Salvador

**DOI:** 10.3390/ijerph16142508

**Published:** 2019-07-14

**Authors:** Priyanka Achalu, Neha Zahid, Dominique N Sherry, Andrew Chang, Karen Sokal-Gutierrez

**Affiliations:** School of Public Health, University of California, Berkeley, CA 94720, USA

**Keywords:** nutrition, oral health, nutrition transition, children’s health, barriers to care, El Salvador

## Abstract

The nutrition transition from traditional diets to processed snacks and sugary beverages has contributed to a higher burden of child malnutrition, obesity, and tooth decay. While child health interventions typically promote nutritious eating, they rarely promote oral health. Mothers’ motivations for child nutrition and oral health practices need to be better understood. A convenience sample of 102 mothers in eight rural Salvadoran communities participated in focus groups addressing child nutrition and oral health. Focus groups were transcribed and coded using qualitative content analysis. Primary themes included generational changes in health environments; health knowledge, attitudes, and practices; and access and barriers to health services. Mothers noted general improvements in awareness of oral hygiene but poorer child oral health, which they attributed to widespread sales of unhealthy snacks and beverages near schools. Distance and cost limited families’ access to dental services. Knowledge gaps included the belief that oral iron supplements cause tooth decay, uncertainty regarding when to start tooth brushing, and until when parents should help children brush. Maternal-child health programs should emphasize the adverse health consequences of feeding young children processed snacks and sugary drinks, and promote dental care access and regulations to ensure health-promoting environments surrounding schools.

## 1. Introduction

Over recent decades, global increases in urbanization, trade liberalization, and food marketing have driven a nutrition transition from traditional, whole-food diets to foods and beverages that are highly-processed and high in sugar, fat, and salt [1,2]. The nutrition transition has had a disproportionately adverse impact on low-income communities in high-income countries and all populations in low- and middle-income countries, contributing to a double burden of malnutrition and obesity [1]. Latin America has experienced increases in dietary fat and sugar consumption, and corresponding decreases in fruit and vegetable consumption [3]. The consumption of nutrient-poor foods and lack of nutrient-rich foods are particularly harmful for young children who are growing and developing rapidly, and are more susceptible to nutrition-related diseases such as malnutrition, obesity, and tooth decay [4,5].

Research on the nutrition transition has emerged from some countries in Latin America, and other countries remain less-studied [6,7,8]. While investments in economic development, education, and maternal-child health programs have led to substantial reductions in rates of child malnutrition in many Latin American countries, improvements in some countries, including El Salvador, have been limited [9]. Studies show that approximately one in four Salvadoran children experience malnutrition [10], placing them at risk for poorer overall health, development, and quality of life [11]. In addition, El Salvador has experienced a drastic increase in child obesity, with the prevalence of overweight children below the age of five rising by 50% over the past decade [12]. Latin American countries, including El Salvador, have a high rate of early childhood caries or tooth decay—affecting over 50% of young children—which can cause chronic infection, inflammation, and mouth pain, and further contribute to malnutrition, poor development and school performance, and reduced quality of life [13,14].

To improve the nutrition and oral health of young children, it is important to understand the current knowledge and practices of mothers and caregivers in order to address their perceived barriers and facilitators in ensuring their children’s health [1,13,14,15]. This qualitative study explores rural Salvadoran mothers’ knowledge, attitudes, and experiences with their children’s nutrition and oral health, with an aim to identify strategies for improvement.

## 2. Materials and Methods

### 2.1. Study Design

This study utilized qualitative methodology with focus groups to assess rural Salvadoran mothers’ knowledge, attitudes, and experiences regarding their children’s nutrition and oral health. Semi-structured focus groups were conducted to engage participants in in-depth discussion of their perceptions, habits, and motivations [16].

### 2.2. Ethical Approval and Community Partnership

The study received ethical approval from the University of California, the Berkeley Committee for Protection of Human Subjects (Institutional Review Board: 2010-06-1655), and the directorship of Asociación Salvadoreña Pro-Salud Rural (ASAPROSAR) (Salvadoran Association for Rural Health), our local Salvadoran partner. ASAPROSAR is a longstanding Salvadoran non-profit, non-governmental organization that collaborates with other local institutions to provide community health and development programs. The maternal-child health and nutrition program incorporates oral health promotion and utilizes trained community health workers.

### 2.3. Study Population

The community health workers recruited a convenience sample of 102 mothers/caregivers of young children across 8 rural, low-resource communities in the Santa Ana region to participate in semi-structured focus groups. Participants completed brief demographic surveys, and means of demographic characteristics were calculated for each community, and for the total sample (Table 1). The communities and participants had generally similar demographic characteristics. On average, mothers/caregivers were 30–31 years of age, low income, had approximately 2–3 children, and had received primary school education.

### 2.4. Data Collection

A total of 22 focus groups were conducted, with an average of 4–5 participants per group. The focus groups were facilitated by two trained native Spanish speakers and assisted by a note-taker. Facilitators first explained to participants the study objectives and procedures, including confidentiality, and obtained signed informed consent from each participant. After participants completed a brief demographic survey, the facilitator led the group discussion based on a semi-structured focus group guide with questions addressing mothers’ understanding of good nutrition and oral health, and the local promoters and barriers to good nutrition and oral health for their children. The questions provided structure to ensure coverage of a standard set of topics with flexibility to foster organic dialogue. All mothers were encouraged to participate in the discussion and elaborate on their comments. The focus group discussions lasted approximately 40 min.

The focus groups were audio-recorded and supplemented by notes. After the focus groups were completed in each community, the facilitators and note-takers debriefed to discuss the key points and determine whether any changes in the wording or delivery of the questions were needed. The debriefing sessions also facilitated recognition of when saturation of themes occurred. Accommodations to conduct additional focus groups were available in the cases where saturation was not reached. The focus group audio recordings and notes were translated from Spanish and transcribed into English.

### 2.5. Qualitative Data Analysis

The focus group transcripts were analyzed using a content analysis approach [17]. The transcripts were coded independently by 3 members of the research team to create a preliminary codebook of 44 distinct themes. Content analysis identified 3 overarching themes with issues related to nutrition and oral health for each major theme. The key response categories were identified by the frequency of the responses and supported by representative quotes from participants.

## 3. Results

Three overarching themes were identified—changes from previous generations; mothers’ knowledge, attitudes and practices; and access and barriers to items and services—with each theme addressing nutrition and oral health issues (Table 2).

### 3.1. Changes from Previous Generations

#### 3.1.1. Nutrition

Most mothers noted that present-day markets offered more variety and greater quantities of food than the markets they remembered from their childhoods. Many also noted the higher prevalence of processed snacks in today’s markets.
“There’s much more variety now… before there was mostly rice and beans.”

Some mothers noted that changing socioeconomic conditions adversely affected their food purchases. Many mothers cited increased food prices and reported greater difficulty purchasing their desired foods. In contrast, some mothers asserted that there is generally less poverty today compared to the Salvadoran civil war era from 1980–1992, and they could now purchase and feed their children a greater quantity of food than before.
“The kids eat more now because there was more poverty when I grew up.”

#### 3.1.2. Oral Health

Most participants reported greater awareness of oral health since their childhood. Many mothers said that schools, the media, and families now devote more attention to oral health and tooth brushing. Many participants mentioned that there is now more access to toothpaste than in the past and that people are brushing more frequently now than previously.
“I don’t remember being told to brush three times a day [by my parents]... but now, with my own kids, I take more care [with their oral health].”
“It’s easier now for the kids to have cleaner teeth because before, when we were growing [up], we almost never had toothpaste at home.”

In contrast, other participants argued that children’s oral health status has declined substantially over recent years. Many mothers cited the increased access to sweet snacks and drinks, resulting in severe tooth decay, which had not been a problem for children in previous generations.
“My kids have it worse [than I did]… their teeth are ruined; I think it’s [from] the sweets.”

### 3.2. Health Knowledge, Attitudes, and Practices

#### 3.2.1. Nutrition

Mothers described how they decided what foods to purchase and prepare at home. They were primarily guided by the Salvadoran dietary traditions, their family members’ food preferences, and the nutritional quality of the food. They also considered financial constraints, the accessibility of items at nearby markets, and ease of storage.
“[We make] whatever can provide good nutrition.”
“I [the mother] decide what food to buy, but I try to make whatever the kids want to eat if they [are craving] something in particular.”

While the mothers primarily prepared traditional, healthy meals at home, they had less control over what their children ate outside of the home. Mothers typically sent their children to school with money to buy food. While they reported telling their children to buy healthy food, they expressed concern that their children often bought non-nutritious snacks and sugary beverages sold in and around the schools. Many mothers were alarmed by the high prevalence of junk food, and by its tastiness, cheap price, and the colorful packaging that attracted their children to buy and consume it. They also believed that the processed snacks were harmful for their children’s overall health and development.
“The presentation of the product is appealing [to the children], even though it’s not healthy.”
“They put in chemicals…and [those] stall [children’s] development.”

#### 3.2.2. Oral Health

Mothers recognized the oral health benefits associated with breastfeeding compared to bottle-feeding their children. One mother stated that feeding her children foods rich in calcium and other nutrients was important for promoting their oral health.
“Breast milk is good for dental health. It has a lot of vitamins that only come from maternal milk.”

Many mothers knew that the processed snacks were harmful to their children’s oral health. Some mothers explained that they tried to limit their children’s consumption of processed snacks to protect their children’s oral health and prevent tooth decay.
“Sweets ruin teeth. When I was a kid, they [the baby teeth] would come out with fingers. Now, I have to pay [the dentist] to get these small pieces out.”
“Don’t eat a lot of junk food because it hurts the teeth and leads to caries.”

Many participants knew that tooth brushing was important to protect their children’s oral health. Most mothers reported reminding their children to brush after meals, but not necessarily helping their children brush. Mothers also stated that access to dental services was important for children’s oral health.
“Brush especially after dinner but also after other meals.”
“Go to dentist for fluoride and fillings.”

Mouth pain was the most commonly cited consequence of tooth decay. Mothers reported that mouth pain caused their child to have difficulty eating, brushing teeth, sleeping, and concentrating in school. The pain adversely affected the quality of life for their child and the entire family. Mothers also believed that poor oral health adversely affected their child’s overall health. Others discussed how poor oral health could negatively affect a child’s physical appearance and overall self-esteem.
“Whenever the kid brushes, he complains of tooth pain—there are four rotten in the front.”
“Yes, [there’s] pain, [and his] teeth got pulled out. They’re affected; we’re affected. They can’t eat, [which means] we have to spend more time with them.”
“Poor oral health causes sickness.”
“If there’s a lot of bacteria in [their] teeth, this can affect the rest of the body and general health if swallowed.”
“It gives them shame to smile in front of people when there’s damage in their teeth.”

There were also some common gaps in knowledge and misconceptions regarding oral health. Mothers debated the proper age to start cleaning their children’s mouth. While many mothers knew that they should start brushing their children’s teeth in infancy when the first baby teeth erupted, few mothers knew that they could use a clean cloth to begin cleaning their children’s gums earlier in infancy. Mothers also debated the age at which children could brush their teeth on their own—some believed that children could brush their teeth once they could hold the brush around age 2, but others asserted that children needed more help brushing until school-age. Many mothers noted that the oral liquid iron supplement prescribed to treat their child’s anemia stained their child’s teeth black and they believed that the iron supplement caused their child’s tooth decay.
“My oral health was better [than my child’s] because I didn’t take iron like my child does now.”

### 3.3. Access and Barriers to Items and Services

#### 3.3.1. Nutrition

Mothers reported that their children primarily accessed junk food by purchasing it themselves, especially at school. Nearly all mothers reported giving each child a daily average allowance of $0.25. When asked how their children spent this money, mothers consistently answered that the children purchased processed snacks and sugar-sweetened beverages, which they generally did not give them at home. Some mothers complained that their children preferred junk food over healthy, traditional foods.
“I give them money and tell them not to eat [junk food], but they buy it anyway.”
“The school has lots of junk food, and it’s sort of inevitable for the kids to buy.”
“The first thing they are going to buy is sweets.”
“The kids would rather eat junk food like churros instead of tortillas.”

Some mothers explained that when extended family members visited, they gave snacks or sweets to the children. There was a general sentiment that giving children sweets was a traditional way to show affection in Salvadoran culture.

Many mothers noted that financial constraints were a barrier to providing good nutrition for their families. Most families had limited income and price was a major concern when grocery shopping. Mothers also mentioned that fresh fruits and vegetables were more likely to spoil compared to processed foods such as instant soup.
“Poverty is complicated—we need to balance purchasing economically with what’s healthy for the kids.”
“I look at price and end up purchasing more convenient [affordable] foods so I don’t spend all of my money.”
“The more nutritious foods tend to be more expensive.”

Distance from markets was also noted as a barrier to good nutrition. The larger supermarkets, with a wider variety of products, were located in urban areas, and it was too time-consuming and costly to access these markets from the rural communities.
“The market in [the city] is far from home. We have to pay 40 cents for the bus; it takes 20 min.”

#### 3.3.2. Oral Health

Mothers reported that toothbrushes and toothpaste were accessible in many local stores. However, they were relatively expensive—the $1 cost for a toothbrush or a tube of toothpaste was approximately one-fifth the average daily income ($5–$6) for a rural family.

Mothers cited multiple barriers to accessing dental health services. These included the distance from their rural villages to clinics located primarily in cities, limited and inconvenient bus schedules, cost of transportation, and risk of losing an entire day of work to complete a visit to the dentist.
“The problem is that the dentist doesn’t come [to our villages], and they aren’t nearby.”
“You need to get [to the clinic] by bus, which costs $2 and takes much of the day to get there. If you don’t catch the bus, you have to pay $6–$7 for a car.”

Mothers additionally explained that once they arrived at the clinic, there was no guarantee that they would receive services.
“The process is not easy or convenient. You need to wait in line and return on different days a lot of the time.”

The high cost of dental care—even for basic treatments such as extractions and fillings—made dental treatment inaccessible for most families. Consequently, most mothers explained that they considered dental care impossible to prioritize or “unnecessary”, unless their children were suffering from unbearable tooth pain.
“The [city] dentist is $10 for extractions, $25 without an appointment, $14 for a filling.”
“We only go [if there is] necessity… because the kids’ teeth don’t hurt, [we don’t go].”
“My kid complains about tooth pain. I would take him to the clinic but I don’t have $5 to cover dental care so we pass with the pain.”

Many mothers agreed that they could better maintain their children’s health if the goods and health services were made more accessible in rural areas through local markets and rural health centers.

## 4. Discussion

This qualitative study of rural Salvadoran mothers’ knowledge, attitudes, and experiences regarding child nutrition and oral health identified three major themes—changes from previous generations, mothers’ knowledge and practices, and access and barriers to items and services—with nutrition and oral health issues for each major theme. Mothers noted both improvements as well as setbacks in the nutrition and oral health environment since their childhoods. On the positive side, a wider variety of foods has become available, mothers continue to prepare their traditional healthy foods at home, and there is an increased emphasis on tooth brushing and access to toothpaste. However, the positive changes may be outweighed by negative factors contributing to poorer nutrition and oral health for children—children today consume unhealthy snacks and beverages daily at school, many children suffer from severe tooth decay and mouth pain, and dental treatment is too costly and inaccessible. In addition, there remain misconceptions about proper tooth brushing and causes of tooth decay.

Overall, mothers demonstrated substantial knowledge regarding healthy nutrition and dental practices, likely enhanced by the local maternal-child health program’s focus on nutrition and oral health promotion in these communities. However, the mothers encountered challenges in translating their knowledge into practice. Regarding nutrition, while mothers knew what food was healthy and tried to serve that at home, they felt unable to resist the marketing, social pressures, and convenience of giving their children money with which they bought unhealthy snacks at school. Regarding oral health, mothers knew the importance of tooth brushing, but had some misconceptions about when and how to help their children brush. In addition, mothers knew the importance of getting their children dental treatment for tooth decay, but the distance and cost made it impossible for them to prioritize treatment, except in the most severe cases. As a result of the caries-causing diet, misconceptions about tooth brushing, and the lack of dental treatment, most children suffered from untreated tooth decay, which could compromise their overall health, nutrition status, and ability to concentrate in school. This reality underscores the importance of addressing oral health within maternal-child health and nutrition services.

Our results are consistent with findings from other studies in Latin America and globally regarding the nutrition transition [4], the widespread practice of feeding young children non-nutritious snacks and sugary beverages [4,7,8], and the adverse consequences on children’s nutrition and oral health [1,4,18,19,20]. Other studies have found that parents have limited knowledge about the contribution of sugary beverages and the baby bottle to early childhood caries [21], the proper age to initiate oral health care for children [22], and that children are not developmentally capable of effectively brushing their teeth and need adult assistance until six to eight years of age [23]. In addition, other studies have identified similar barriers to accessing dental treatment services, including families’ financial constraints [24,25] and limited dental services in rural areas [26].

These findings support the need to incorporate a socio-ecological approach to develop interventions that help families put their knowledge into practice through socioeconomic, cultural, and environmental supports for child nutrition and oral health [27,28,29]. Policy changes are needed to make healthy choices accessible and affordable for families, e.g., by restricting vendors selling unhealthy foods and ensuring access to free and low-cost healthy foods and clean water in and around schools. In 2014, every Latin American country signed a Pan American Health Organization agreement to prevent childhood obesity through enacting a variety of policies, including supporting nutritious school food programs and prohibiting non-nutritious snacks and beverages from schools [30]. New legislation in El Salvador, such as the National Food and Nutrition Security Policy for 2018–2028, has also been passed to improve food availability and price stability to better guarantee access to healthier food options [31]. However, while countries have made substantial progress, better enforcement of these regulations is needed. Children in school environments are still widely targeted for the sale of unhealthy products, and policies supporting access to fresh, local farm products are also not well enforced [32,33,34,35]. In addition, rural families need access to oral health promotion, linked with overall health and nutrition promotion, and affordable preventive and curative dental care through rural health posts and maternal-child health and nutrition programs where they can receive prenatal care and child immunizations [15,18,36,37]. Further research is needed to explore additional factors, such as social norms and aesthetics, that may motivate families’ oral health behaviors. Engaging rural community members in the design and implementation of interventions can help ensure that their needs and priorities are addressed [38,39].

The strengths of this study include a substantial sample size, which allowed for thematic saturation and strengthened the internal validity of the results. Additionally, this study builds on existing quantitative findings [14] to elucidate the “whys and hows” behind the health outcomes associated with the nutrition transition. Study limitations include recall bias when mothers discussed events from their childhoods, and possible response bias to shape answers to please the facilitators. Qualitative data alone, without supporting quantitative data, has inherent limitations regarding the comprehensiveness of the data [16]. Finally, since the study population is a convenience sample from rural communities surrounding Santa Ana, the results may not be generalizable to other rural populations or across different regions of El Salvador.

## 5. Conclusions

This sample of rural Salvadoran mothers/caregivers demonstrated basic knowledge regarding child nutrition and oral health. However, they experienced substantial barriers to putting their knowledge into practice—primarily the widespread sale of low-cost, non-nutritious snacks and sugary beverages in and around schools, and social pressure to give their children money to buy daily treats. In addition, they experienced significant barriers to accessing dental services including distance, transportation, and cost. Maternal-child health programs should incorporate oral health promotion and provide families clear messages about the adverse impact of processed snacks and sugary beverages on children’s nutrition and oral health. In addition, they should help parents develop skills to put healthy nutrition and oral health behaviors into practice, and advocate for access to dental care and the enforcement of local and national policies to ensure health-promoting environments for children in and around schools.

## Figures and Tables

**Table 1 ijerph-16-02508-t001:** Participant Demographic Information.

Participant Characteristics	Community A ^1^	Community B ^1^	Community C ^1^	Community D ^1^	Community E ^1^	Community F ^1^	Community G ^1^	Community H ^1^
Number of participants (n = 102)	11	14	15	10	19	7	11	15
Number of focus groups (n = 22)	3	3	3	2	5	2	2	2
Average age (mean = 30.7 years)	30.8	25.7	28.2	28.0	30.7	35.7	34.9	32.1
Average daily family income (mean = $5.6)	NA	NA	$8	$5	$5	NA	$5	$5
Average education grade level (mean = 5th grade)	4.6	4.7	5.2	4.3	4.7	4.2	5.4	5.4
Average number of children per mother (mean = 3.2 children)	4.2	2.6	2.5	2.8	3.4	2.9	4.5	3.4
Average household size (mean = 5.4 members)	5.9	5.7	5.4	4.9	5.7	4.4	7.3	4.5
Average child age (mean = 7.4 years)	12.3	3.5	3.6	5.0	7.2	7.9	12.7	7.4

^1^ The names of each community are represented by letters A–H to protect the participants’ anonymity.

**Table 2 ijerph-16-02508-t002:** Summary of major themes and sub-themes.

Health Issue	Changes from Previous Generations	Health Knowledge, Attitudes and Practices	Access and Barriers to Healthy Items and Services
NUTRITION	Increased variety and quantity of foods Increased prevalence of junk food	Knowledge about what foods/drinks are healthy and unhealthy Traditional foods served at home Poverty limits food purchases Money given to children used to buy junk food at school	Children have access to low-cost, attractive, and tasty junk food at school Family members bring children sweets Limited access to fresh food due to challenges with transportation, time, cost, and spoilage
ORAL HEALTH	Increased awareness of oral hygiene Increased availability of toothbrushes and toothpaste Increased prevalence and severity of childhood caries due to frequent consumption of sweet snacks and drinks	Knowledge that sweets cause caries, and breastfeeding, limiting sweets, brushing teeth, and dental care can promote oral health Knowledge that caries cause mouth pain and poor general health Uncertainty about how/when to help children brush, and misconceptions about iron causing caries	Challenges to buy toothbrushes and toothpaste on low income Limited access to dental treatment due to transportation, time, and cost

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
