# Peer review of "A Qualitative Study of Child Nutrition and Oral Health in El Salvador"

_ijerph, 2019, doi:10.3390/ijerph16142508_

Round 1

Reviewer 1 Report

This paper is a study of child nutrition and aural health in El Salvador. The effects of nutrition transition, economic development, mother's knowledge and practices on oral hygiene were examined.

The focus is interesting, however this is not an article, but a summary of a questionnaire. However, this is not an article, but an enumeration of answers to the questionnaire. A non random selection of answers is highly subjective. In this case, statistical processing must be performed to objectively present the data. 

You described it as "Mothers noted both positive and negative changes in the nutrition and oral health environment since their childhoods." in L240.

What percentage is positive? and negative? What are the regional differences? What percentage of these causes are toothpaste, brushing, knowledge, food, or treatment? Most of the references you cited are also statistically processed. Use the references as a guide for statistical processing.

Author Response

Please see the attached file's section for Reviewer 1. 

Reviewer 2 Report

The paper is interesting but it needs some major revsions.

Materials and Methods:

- the Ethical Committee protocol number is missing (line 59);

- the description of the sample is not clear, it needs to be improved;

- the paper is mssing in description of statistical analysis.

Results:

- this section is too long and the results are not well described, it is a not scientific way to describe results

Discussion

- this section can be improved.

Some references are too old.

-  Methods is lack of statistica anlysis and Ethical committee protocol number, so the esperiment can’t be replied without these details

- Results are to rewritten in a different style, more scientific without citino all the words of the partecipanti

- Discussion is a good section but it seems to be poor, it can be improved

Author Response

Please see the attached file's section with responses for Reviewer 2. 

Reviewer 3 Report

The authors present a nice, well written qualitative study about the perspectives of mothers in El Salvador regarding the quality of oral health care, changes in generational habits, and access to healthy food choices for their children.  It was interesting to read the contrasting perspectives among the mothers and also have their words quoted directly.  This study was well done.

Please check the attached file for detailed comments.

Author Response

Please see the attached file's section with responses for Reviewer 3. 

Round 2

Reviewer 1 Report

I personally would like to see more statistical analysis, but this article is better than last time.

Reviewer 2 Report

THE PAPER CAN BE ACCEPTED